# REPRESENTATION COLLAPSING PROBLEMS IN VECTOR QUANTIZATION

**Wenhao Zhao** *
National University of Singapore
wenhaozhao@u.nus.edu

**Qiran Zou** *
National University of Singapore
qiranzou@gmail.com

**Rushi Shah**
Indian Institute of Technology Jodhpur
shah.15@iitj.ac.in

**Dianbo Liu**[†]
National University of Singapore
dianbo@nus.edu.sg

## ABSTRACT

Vector quantization is a technique in machine learning that discretizes continuous representations into a set of discrete vectors. It is widely employed in tokenizing data representations for large language models, diffusion models, and other generative models. Despite its prevalence, the characteristics and behaviors of vector quantization in generative models remain largely underexplored. In this study, we systematically investigate the issue of collapses in vector quantization, where collapsed representations are observed across discrete codebook tokens and continuous latent embeddings. By leveraging both synthetic and real datasets, we identify the severity of each type of collapses and the conditions leading to these collapses, as well as analyze their underlying causes. Accordingly, we propose potential solutions aimed at mitigating these collapses. To the best of our knowledge, this is the first comprehensive study examining representation collapsing problems in vector quantization.

## 1 INTRODUCTION

Vector Quantization (VQ) technique (Gray, 1984), which discretizes data representations, has become a widely used tool in different aspects of machine learning, including but not limited to tokenization, information bottlenecking, and latent representation. VQ efficiently compresses and represents data by dividing the continuous data space into a limited number of regions and using representative points within these regions to approximate the original data, which converts the original continuous representation into a discrete form. This method not only enhances computational efficiency but also increases model robustness by reducing sensitivity to minor variations in data.

VQ offers a broad spectrum of applications across various domains, especially by tokenizing continuous representation into discrete space. In the field of computer vision, VQ-based tokenization supports high-quality image generation by compressing images into low-dimensional discrete tokens (Van Den Oord et al., 2017; Chang et al., 2023; Esser et al., 2021; Ramesh et al., 2021). Moreover, in audio processing, VQ tokenization enhances the naturalness and expressiveness of synthesized sounds (Chung et al., 2020; Dhariwal et al., 2020). Another significant advantage of VQ-based tokenization is its support for multimodal learning, which simplifies the learning and integration of features across different data sources by mapping them into the same discrete space. This capability not only strengthens the model's ability to handle complex data but also enhances its flexibility and efficiency in multi-domain applications (Ramesh et al., 2021).

Despite the prevalence and potentials of VQ across various fields in machine learning, challenges involving quantization errors and insufficient representational capacity have been observed that limit its further development and application. For instance, when applying VQ to image generation tasks,

---

*Equal contribution
[†]Corresponding author

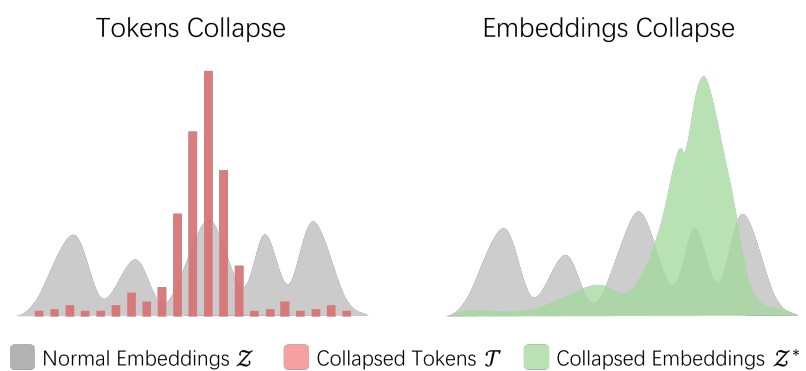

Figure 1: **Representation collapse types in vector quantization.** On the left, *Tokens Collapse* is illustrated, where a subset of tokens (shown in red) collapses, leaving fewer codes for other peaks and losing diversity compared to normal embeddings (in grey). On the right, *Embeddings Collapse* is shown, where a large portion of the embedding space (in green) collapses into a limited set of representations losing out on important information present in other modes. Both phenomena lead to a degradation in the quality of learned representations.

these challenges may cause the generated results to be overly uniform, lacking the necessary diversity and precision, which impacts the practicality and scalability of VQ technique.

In this study, we identify representation collapsing issues associated with VQ, that can potentially lead to decreased output diversity, uneven quality, loss of certain modes compared to the original data, and the generation of distorted modes. These collapsing issues can be categorized into two levels: **a) Tokens Collapse**: A disproportionate number of tokens are allocated to only a few embeddings, leaving other embeddings with few codes. As shown in Fig. 1 (left), a large number of tokens are concentrated at the central peak of embeddings distribution, rather than being proportionally and evenly distributed across the peaks according to the embedding distribution. This results in a lack of sufficient tokens to represent certain data modes, leading to poor representation and a loss of diversity. **b) Embeddings Collapse**: We observe that insufficient encoder parameters can also lead to inadequate representation during VQ process. It results in the outputs of input data from different categories clustering together after being processed by the encoder, hindering the learning of discrete representations during the VQ process, eventually leading to embeddings collapse. As shown in Fig. 1 (right), the distribution of collapsed embeddings from an encoder with insufficient parameters, compared to the distribution of normal embeddings from an adequately parameterized encoder, has fewer modes (peak numbers). This poses a challenge to VQ in learning distinctive and meaningful discrete representations.

To address these issues, we conducted detailed experiments to analyze the causes of each type of collapse and proposed corresponding solutions. First, for tokens collapse, we found that the commonly used initialization of the tokens, which is based on the output of an untrained encoder, can lead to this issue. The untrained encoder does not yet understand the semantics of the data, resulting in embeddings lacking distinction and clustering together. Fig. 2 shows the squeezed embeddings from the untrained encoder's initialization. This clustered initial distribution is deviated from the ideal, dispersed distribution, which makes it difficult to effectively train the tokens and eventually leads to tokens collapse. To counter this, we propose a simple yet highly effective strategy: pretrain without VQ, then fine-tune with VQ. This approach allows the tokens to be initialized based on the semantics of the data as understood by the pretrained encoder, thereby reducing resistance in VQ training. Our solution exhibits a trend of performance improvement as the number of tokens increases, surpassing the baseline by leveraging the benefits of increasing codebook size. Furthermore, for embeddings collapse, since we found that insufficient encoder parameters could result in weakening the encoder's perceptual abilities, it is noteworthy that increasing the encoder's parameter count can mitigate this issue in practical applications of vector quantization.

In summary, our work aims to systematically investigate the causes of the aforementioned two levels of collapses and to propose appropriate solutions accordingly. The contributions are three-fold:

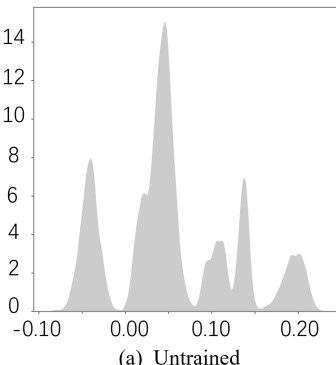 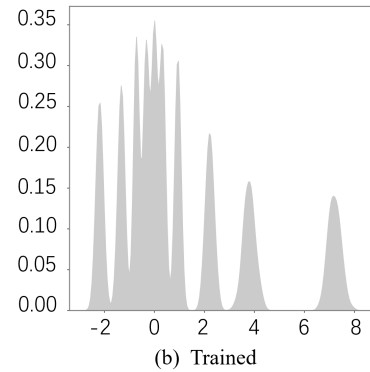

(a) Untrained (b) Trained

Figure 2: **Distribution of untrained and trained encoder's output.** (a) Untrained encoder's output has fewer peaks than 10 peaks of input and clusters around a relatively small range. (b) Trained encoder's output displays 10 peaks which is the same as the input.

- Our research systematically identifies two types of collapsed representation in vector quantization: tokens collapse and embeddings collapse.
- We conducted extensive experiments to thoroughly analyze the causes of the two collapses, elucidating how these factors contribute to collapses and ultimately hinder VQ performance.
- Based on our analyses, we proposed potential solutions to address each type of collapse, shedding light on further improvements and applications of VQ.

## 2 PRELIMINARIES

### 2.1 VECTOR QUANTIZATION

We define the VQ-VAE as following: an encoder $E_\theta$, a decoder $D_\theta$, and a set of tokens $\mathcal{T} = \{t_1, t_2, \ldots, t_S\}$. The token set $\mathcal{T}$ constitutes the codebook, which is utilized to store the discretized representations. The encoder is responsible for mapping the raw data $X = \{x_1, x_2, \ldots, x_N\}$ to a set of continuous representations $\mathcal{Z} = E_\theta(X)$, where $\mathcal{Z} = \{z_1, z_2, \ldots, z_N\}$. And the decoder reconstructs the data $X' = D_\theta(\hat{Z})$ based on the set of discretized representations $\hat{Z}$, where $\hat{Z} = \{\hat{z}_1, \hat{z}_2, \ldots, \hat{z}_N\}$. The process of tokenizing a continuous representation $z_j$ to discrete representation $\hat{z}_j$ is as following:

$$\hat{z}_j = \arg \min_{t_k \in \mathcal{T}} \|z_j - t_k\|, \tag{1}$$

where $t_k$ is a token in token set $\mathcal{T}$ and $k$ is the index of $t_k$ in the codebook. This quantization process is achieved by finding the nearest token $t_k$ in $\mathcal{T}$. In addition, to differentiate from "codebook collapsing", which denotes the problem of low utilization rates of codes within the codebook, we employ another widely used term "token" instead of "code" to represent the vectors of discrete representations in the codebook.

### 2.2 EXPERIMENT SETTINGS

We conduct experiments on synthetic and real-world data to show the two types of collapses and further investigate the reasons as well as validate the effectiveness of our proposed solutions. Our synthetic dataset comprises ten classes, each containing an equal number of samples. The uniform class distribution aims to highlight disproportionate tokens distribution, making the collapse phenomenon more observable. For experiments on synthetic dataset, we generate data with different dimensions to investigate the collapses under different complexity of data. In addition, we adopt t-SNE decomposition to visualize synthetic data exceeding three dimensions. Furthermore, we use CIFAR-10 dataset for experiments on real-world data. The codebook usage and model performance are evaluated by perplexity and MSE respectively.

Throughout the experiment, we adopt the K-means based update method for codebook (Van Den Oord et al., 2017), which updates the tokens by finding the cluster centers of the encoder

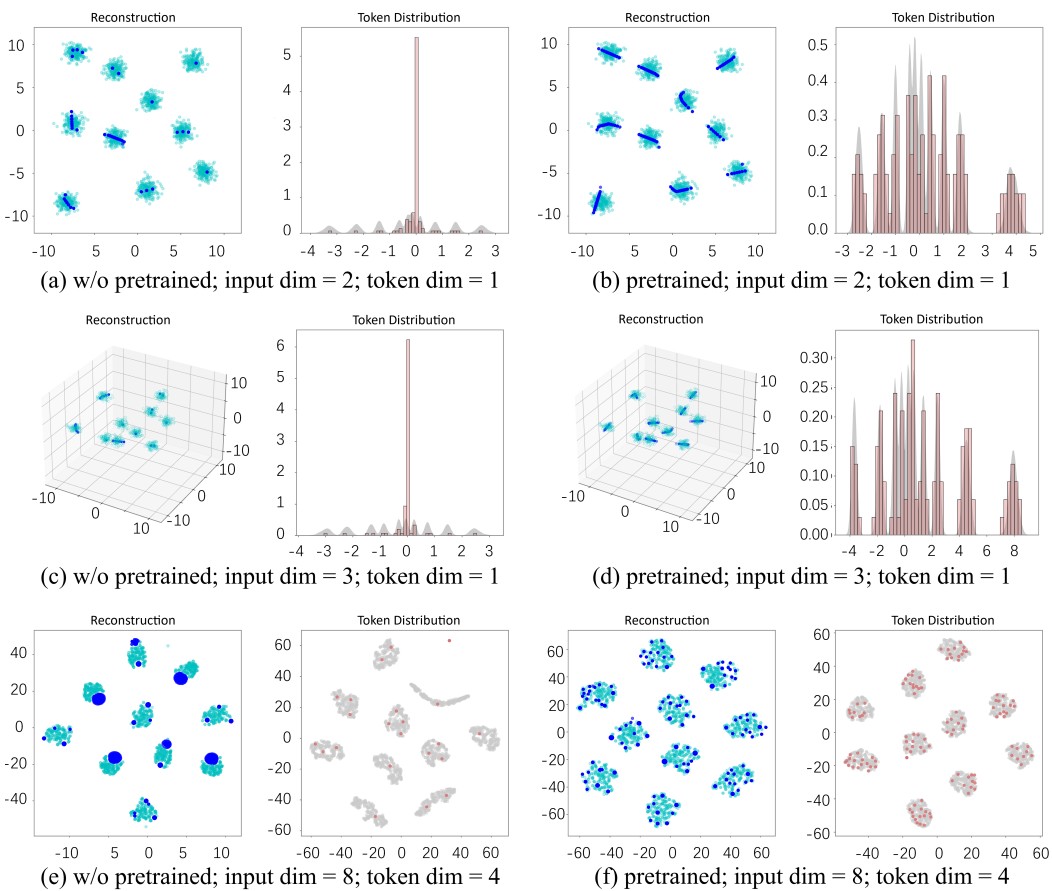

(a) w/o pretrained; input dim = 2; token dim = 1

(b) pretrained; input dim = 2; token dim = 1

(c) w/o pretrained; input dim = 3; token dim = 1

(d) pretrained; input dim = 3; token dim = 1

(e) w/o pretrained; input dim = 8; token dim = 4

(f) pretrained; input dim = 8; token dim = 4

Figure 3: **Tokens collapse and results of our pretraining solution on synthetic data**. The comparison between results with and without our pretraining solution demonstrates that the untrained encoder is able to result in tokens collapse and our pretraining solution is effective.

outputs. Moreover, we also use K-means initialization method proposed by SoundStream (Zeghidour et al., 2021). It applies K-means to get centroids of encoder output, which are used to initialize codebook. Although this method achieved some success, we find that initializing with an untrained encoder could lead to tokens collapse.

## 3 COLLAPSING PROBLEMS AND SOLUTIONS

### 3.1 TOKENS COLLAPSE

Tokens collapse manifests as a disproportionate concentration of the tokens distribution around a subset of the encoder output embeddings. This collapse results in a poor representation since the ideal scenario would involve a fitting distribution of tokens and embeddings that aligns more closely. As demonstrated in Fig. 3 (a), (c), and (e), embeddings assigned only a few tokens exhibit severely insufficient representations, which leads to the corresponding reconstructed distribution being narrower, compared to the reconstruction of more equitable tokens distribution (Fig. 3 (b), (d), and (f)), indicating the loss of diversity.

What causes tokens to collapse? We observe that during the initialization of codebook, tokens often cluster within a very small range, leading to early tokens collapse. As shown in Fig. 2, compared to trained encoder (Fig. 2 (b)), the distribution of untrained encoder's outputs cluster between $-0.10$ to $0.20$ (Fig. 2 (a)) and has only 5 obvious peaks while the input dataset contains 10. This phenomenon

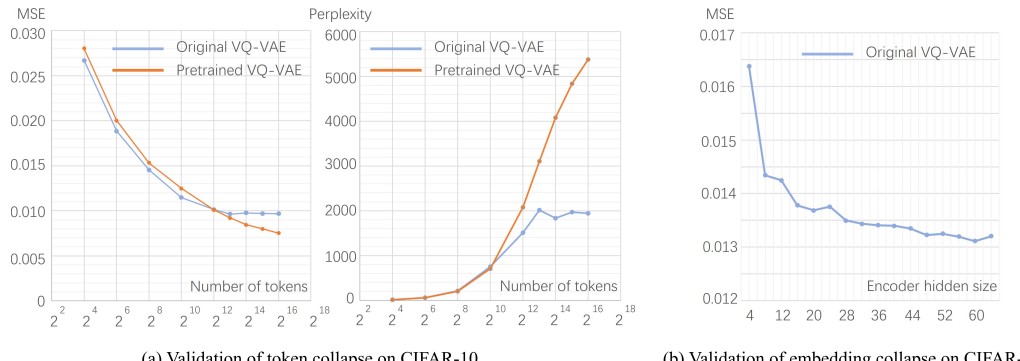

(a) Validation of token collapse on CIFAR-10       (b) Validation of embedding collapse on CIFAR-10

Figure 4: **(a)** As the total number of tokens increases, the MSE and perplexity for pretrained VQ-VAE and original VQ-VAE models reveal distinct behaviors. From $2^{12}$, the original VQ-VAE start to suffer from severe token collapse due to dense tokens, causing MSE and perplexity to stagnate. Conversely, the pretrained VQ-VAE addresses this issue, resulting in continually decreasing MSE and increasing perplexity. **(b)** The hidden size of encoder significantly influences the performance of VQ-VAE. Insufficient parameters prominently results in an embedding collapse problem.

is primarily due to using outputs from an untrained encoder for initialization: an untrained encoder fails to understand the input data, resulting in most data being encoded into similar embeddings.

Building on these observations, we hypothesize that if tokens are initialized based on encoder that has learned semantic distinctions and its output embeddings are dispersed, it would enhance the semantic distinction among tokens and thus control tokens collapse. Consequently, we propose a straightforward yet effective method to mitigate tokens collapse: pretrain without VQ, then fine-tune with VQ. It first trains an autoencoder, and then trains the VQ-VAE initialized with the weight of the autoencoder trained at the first stage. Pretraining the encoder allows it to discern differences in input data, resulting in more distinctly spaced embeddings, providing a robust foundation for initializing the tokens, as demonstrated in Fig. 2 (b).

### 3.1.1 EXPERIMENTS ON SYNTHETIC DATASET

To validate our hypothesis regarding the causes of tokens collapse and the effectiveness of our proposed solution, we conducted ablation studies with and without pertaining under different input data and token dimensions. In Fig. 3, settings (a), (c), and (e) represent commonly used methods without pertaining, compared to the settings (b), (d), and (f) which adopt our pretraining solution. In addition, results under different input and token dimensions are included.

The experiment results under 2-dim synthetic dataset is shown in Fig. 3, (a) and (b). Comparisons between (a) and (b) demonstrates that, with our pretraining solution, the distribution of tokens is more uniform and the reconstruction distribution covers closer to the original data, indicating the effectiveness of our solution for tokens collapse.

Even at higher input data dimensions and increased token dimensions, we observed that tokens collapse problem persists, and our solution maintains high effectiveness. As shown in Fig. 3, in the comparison between (c) and (d) for 3-dimensional input data, and between (e) and (f) for 8-dimensional input data, the issue of tokens collapse can be observed, as well as the problem where the reconstructed data fails to adequately cover the original input data. It also shows that these issues were effectively mitigated when adopting our solution.

### 3.1.2 EXPERIMENTS ON REAL-WORLD DATASET

To validate that our solution can address tokens collapse under real data conditions, we conducted corresponding experiments on CIFAR-10 dataset. Additionally, we hypothesize that given a fixed dimensionality of the representation space, an increase in the number of tokens tends to facilitate their clustering, thereby making tokens collapse more pronounced. Under these conditions, it is likely that the benefits of our solution become more evident. Therefore, we evaluated the performance of our solution compared to the absence of it across varying token quantities.

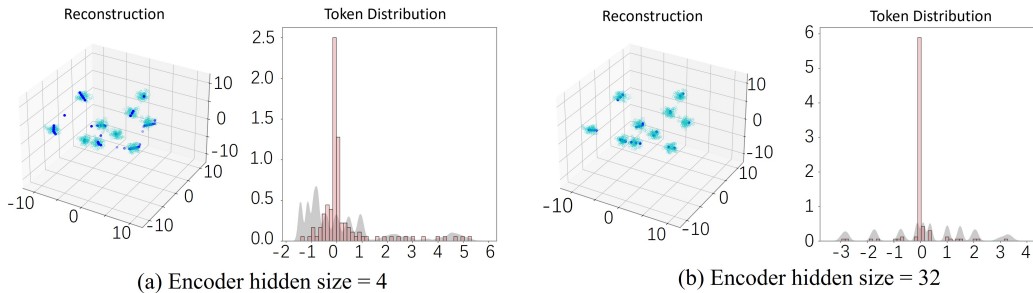

(a) Encoder hidden size = 4            (b) Encoder hidden size = 32

Figure 5: **Embeddings collapse problem on synthetic data.** Compared to the encoder with high capacity (hidden size 32), encoder with low capacity (hidden size 4) exhibits embeddings collapse.

As shown in Fig. 4 (a) (left), it is observable that at lower token numbers, our solution exhibits a slight performance disadvantage compared to the absence of it. However, as the token number increases, our method maintains a trend of performance improvement, whereas the original VQ-VAE encounters bottlenecks at 4096 tokens, struggling to achieve further enhancements. Eventually, with the increase in token number, the performance of our method significantly surpasses that of the original approach with lower MSE and higher codebook perplexity.

One possible reason our method underperforms the original approach at low token numbers is the gap between the discrete representations learned during pretraining and the continuous representations during finetuning, which poses challenges to the VQ learning process. However, this negative impact is outweighed by the benefits of our solution as the codebook size increases. Overall, our approach not only addresses tokens collapse but also unleashes the potential of VQ, further leveraging the benefits of a large codebook. Additionally, exploring how to mitigate the performance gap when the token number is low remains a worthy avenue for further investigation.

## 3.2 EMBEDDINGS COLLAPSE

As discussed in the introduction, insufficient parameters of the encoder could lead to embeddings collapse during the VQ process. This observation suggests the importance of appropriately sizing models and offers insights and rationale for balancing model complexity with computational efficiency to optimize comprehensive performance and prevent collapses in VQ applications.

We conducted an in-depth investigation and validation of this issue on both synthetic dataset and the real-world dataset. We adjusted the encoder's capacity by reducing the parameters of the standard VQ-VAE's encoder (without using our pretraining solution). The overall results demonstrate that insufficient encoder capacity is able to result in the embeddings collapse problem.

Experiment results on synthetic data, as shown in Fig. 5 (b), show that with an encoder hidden size of 32, the distribution of embeddings exhibits clear differentiation, which aids in the learning of discrete representations (tokens). Consequently, the reconstruction results do not contain any outputs that fall outside the distribution of the original data. However, when the encoder's capacity is insufficient (hidden size=4), as shown in Fig. 5 (a), the output embeddings from the encoder tend to have most of their peaks merged together, and parts of the tokens distribution lie outside the embeddings distribution. This leads to the embeddings collapse issue that reconstruction contains erroneous results which fall outside the original input data distribution.

We further investigate embeddings collapse problem on real-world dataset, CIFAR-10. We adjust encoder's capacity by changing the hidden layer size of the encoder. As shown in Fig. 4 (b), it can be observed that decreasing the size of the encoder undermines the reconstruction performance with increased MSE, indicating that insufficient capacity of the encoder can lead to embedding collapse on real-world data.

Therefore, for specific applications of VQ, a guiding suggestion is to ensure the model has sufficient capacity, such as by adding an additional network layer before quantization.

# 4 RELATED WORKS

Vector Quantization is foundational in data compression and signal processing per Shannon's rate-distortion theory (Gersho & Gray, 2012) (Cover, 1999), traditionally relied on methods like K-means clustering (Macqueen, 1967) but faced high complexity with high-dimensional data (Le Tan et al., 2018). To mitigate this challenge, DeepVQ (Le Tan et al., 2018) improved efficiency by mapping data to lower-dimensional latent spaces before quantization. Moreover, (Van Den Oord et al., 2017) proposed VQ-VAE which integrates VQ with variational autoencoders, using a straight-through estimator (Bengio et al., 2013) to handle discrete variables. To refine VQ methods for improved performance, variants such as Residual Quantization Lee et al. (2022), Product Quantization (Chen et al., 2020), and Soft Convex Quantization (Gautam et al., 2023) further enhanced representation capacity and efficiency. Recent advances incorporate attention mechanisms and transformer architectures (Vaswani, 2017) (Yu et al., 2021) to dynamically select codebooks and capture global data dependencies. Recent works also explore per-channel codebooks (Hsu et al., 2024) and neural network variants of residual quantization (Huijben et al., 2024) to predict specialized codebooks, enhancing the model's expressive power.

VQ has been extensively applied across various domains. In natural language processing, VQ facilitates sequence modeling (Kaiser et al., 2018) enhancing tasks such as language modeling and machine translation. In computer vision, VQ has significantly advanced image generation and compression techniques (Esser et al., 2021). Similarly, in audio processing, VQ techniques have captured complex temporal dependencies (Dhariwal et al., 2020). Furthermore, in multimodal applications, VQ supports the integration of different data types through shared discrete representations (Ramesh et al., 2021).

Despite these advancements, VQ methods encounter challenges that restrict their broader application, including but not limited to codebook collapse, training instability, and computational overhead. Extensive research has been conducted on solving the codebook collapse problem, where only a subset of tokens are utilized leading to inefficient representation usage and reduced diversity in outputs, by reducing token dimension (Yu et al., 2021), orthogonal regularization loss (Shin et al., 2023), multi-headed VQ (Mama et al., 2021), finite scalar quantization (Mentzer et al., 2023), and Lookup Free Quantization (Yu et al., 2023). Recent methods like (Goswami et al., 2024) and (Baykal et al., 2024) also strive to enhance tokens utilization efficiency. However, beyond the widely recognized issue of codebook collapse, our work identifies, investigates, and proposes potential solutions for collapses of tokens and reconstruction, which pose serious challenges to VQ learning and merit attention.

# 5 METHODS

## 5.1 VECTOR QUANTIZATION

We employ VQ-VAE to conduct vector quantization to investigate the collapsing issues. For codebook initialization, we adopt the widely used K-means initialization strategy (Zeghidour et al., 2021). It uses the encoder output $\mathcal{Z} = \{z_1, z_2, \ldots, z_N\}$ and perform K-means algorithm to initialize the tokens $\mathcal{T} = \{t_1, t_2, \ldots, t_S\}$, where $N$ is the number of encoder output and $S$ is the number of tokens. The initialization aims to minimize the total distance from each vector $z_j$ to its nearest token $t_k$. The optimizing function is shown in equation 2,

$$\min \sum_{j=1}^{N} \sum_{k=1}^{S} r_{jk} \|z_j - t_k\|^2, \tag{2}$$

where $r_{jk} = 1$ if $z_j$ is assigned to cluster center $t_k$, otherwise $r_{jk} = 0$ .

Moreover, the widely adopted VQ-VAE optimization objective comprises reconstruction loss $\mathcal{L}_{\text{recon}} = \|X - \hat{X}\|^2$, codebook loss $\mathcal{L}_{\text{commit}} = \frac{1}{N} \sum_{j=1}^{N} \|sg(z_j) - t_{jk}\|^2$, and commitment loss $\mathcal{L}_{\text{commit}} = \frac{1}{N} \sum_{j=1}^{N} \|z_j - sg(t_{jk})\|^2$, where $sg(\cdot)$ denotes the stop-gradient operation and $t_{jk}$ means the token selected by $z_j$. Additionally, we adopt the statistical approach (Van Den Oord et al., 2017) to update the codebook instead of the codebook loss term. Specifically, each encoder output $z_j$ is assigned to subsets $\mathcal{Z}_k = \{z_{1k}, z_{2k}, z_{3k}, \ldots\}$ based on nearest neighbor queries within the set

$\mathcal{T} = \{t_1, t_2, \ldots, t_S\}$, where $\mathcal{Z}_k$ comprises embeddings closest to $t_k$. Each $t_k$ serves as the cluster center for $\mathcal{Z}_k$, which receives updates accordingly. However, due to the necessity of training models using minibatches, exponential moving averages (EMA) are utilized to accommodate batch updates. Under EMA framework, the sum and size of $\mathcal{Z}_k$ as $m_k$ and $l_k$. The statistical update process is formalized by the following equation 3 to 5:

$$M_k^{(o)} = \gamma M_k^{(o-1)} + (1 - \gamma)m_k^{(o)}, \tag{3}$$

$$L^{(o)} = \gamma L_k^{(o-1)} + (1 - \gamma)l_k^{(o)}, \tag{4}$$

$$t_k^{(o)} = \frac{M^{(o)}}{L^{(o)}}, \tag{5}$$

where $o$ is the index of iteration and $M$ as well as $L$ are respectively recordings of $m$ and $l$. The $\gamma$ is the decay rate to control the speed of update.

## 5.2 IMPLEMENTATION DETAILS

**Datasets** Our synthetic dataset includes 10 clusters, each with 1,000 data points sampled from a Gaussian distribution and standardized using Standard Scaler. We construct different synthetic data with different dimension (2, 3, 4, and 8). We use CIFAR-10, which consists of 60,000 32x32 color images divided into 10 classes, to investigate our proposed collapses and validate corresponding solutions on real-world data.

**VQ-VAE** For our synthetic dataset, our VQ-VAE comprises an MLP-based encoder/decoder with three linear layers and uses the ReLU activation function. Training is facilitated by the AdamW optimizer, with a learning rate of 0.001. Additional specifications include a codebook size of 128, a hidden dimension of 32, a batch size of 256, a beta of 0.25, and a decay rate ($\gamma$ for EMA) of 0.9. For CIFAR-10, our VQ-VAE adopts downsampling using a CNN with a downsample channel of 128, and the model includes two residual blocks with a hidden channel size of 64. The codebook size is set at 512 with a token dimension of 64. The learning rate is 3e-4, using the Adam optimizer with amsgrad set to true. The beta is 0.25 and the decay rate is 0.99.

**Tokens Collapse** Our solution for tokens collapse comprises pretraining an autoencoder, then fine-tuning a VQ-VAE initialized with the pretrained autoencoder's weight. On the synthetic dataset, the autoencoder is trained for 100 epochs. The fine-tuned VQ-VAE and the original VQ-VAE are trained for 100 and 200 epochs respectively. We also explore various data dimensions (2, 3, and 8) with corresponding token sizes of 1, 1, and 4. For experiments on CIFAR-10, the codebook size is varied from 16 to 65,536, with embedding and token adopting the size of 32. And we pretrain AE for 150 epochs and fine-tune the VQ-VAE for 150 epochs.

**Embeddings Collapse** The hidden size of the decoder is maintained while the encoder's hidden size is reduced to 8. These experiments are repeated three times with different seeds over 200 epochs. For the real dataset, the channel size of downsampling part is gradually reduced from 64 to 2 across 300 epochs.

## 6 CONCLUSION

In this work, we provide an in-depth examination of representation collapsing problems in vector quantization, identifying two levels of collapses, including tokens and embeddings collapses, and investigate their detrimental impacts on VQ. Through detailed analysis with both synthetic and real-world data, we pinpoint the causes of these collapses and introduce potential solutions, which shed light on further improvements and applications of VQ. While our work systematically explores these collapses and offers potential solutions, growth opportunities abound. For example, the transition from continuous to discrete representations in our solution, when pretraining without VQ followed by fine-tuning with VQ, introduces a representation gap that needs addressing. In the future, we plan to further explore the impact and solutions for these collapses in generative models, such as LLMs and diffusion models.

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
