# OpenReview forum: "Representation Collapsing Problems  in Vector Quantization"
_NeurIPS.cc/2024/Workshop/SafeGenAi — SafeGenAi Poster_

### Official Review · Reviewer_p8iP · 2024-10-09

**Rating:** 7
**Confidence:** 4

**Review:**

Summary: The paper identifies two main challenges in Vector Quantization (VQ): Tokens Collapse and Embeddings Collapse. To address these, it proposes several effective solutions, including pretraining without VQ followed by fine-tuning with VQ, increasing the codebook size, and optimizing encoder parameters. These strategies aim to enhance VQ's performance and improve data representation.

Strengths:
1. The paper effectively pinpoints the causes of both tokens and embeddings collapse and provides well-thought-out potential solutions. The logic is clear and coherent, making it easy to follow the reasoning.
2. The use of both synthetic and real-world data to validate their claims adds robustness and credibility to their proposed solutions. The results from the experiments convincingly demonstrate the effectiveness of their approach.

Weaknesses:
1.  While the paper proposes effective solutions (pretraining and increasing encoder capacity), it might benefit from exploring a broader range of alternatives or comparing them with other SOTA techniques to provide a more comprehensive evaluation.
2. The reliance on pretraining without VQ to solve token collapse is an important step but may limit flexibility. Discussing how the method might generalize to cases where pretraining is not feasible or other limitations is necessary.
3. Empirical results in a broader context of VQ applications (e.g., how these collapses affect different domains like NLP, audio processing, etc.) could provide greater insight into the generalizability of the solutions across various fields, as discussed in the conclusion.

---

### Official Review · Reviewer_GFvE · 2024-10-09
**Great independent ideas and design, combined experiments are recommended**

**Rating:** 6
**Confidence:** 5

**Review:**

This paper addresses a crucial and underexplored issue in vector quantization (VQ), specifically the phenomenon of representation collapse in both token distributions and latent embeddings. The authors systematically investigate two types of collapsing problems: Tokens Collapse, where a disproportionate number of tokens are allocated to a few embeddings, leading to reduced diversity in learned representations, and Embeddings Collapse, where insufficient encoder parameters result in poor separation of embeddings. The authors propose two key solutions: pretraining without VQ followed by fine-tuning with VQ to mitigate Tokens Collapse, and increasing encoder capacity to address Embeddings Collapse. Experiments conducted on both synthetic and real-world datasets (CIFAR-10) demonstrate the effectiveness of these solutions.

There are the some improvements the authors could make:
1. Combined Collapse Mitigation:
One limitation of the paper is that the solutions for Tokens Collapse and Embeddings Collapse are tested independently. In practical applications, both forms of collapse may occur simultaneously. It would be beneficial to explore whether a combined approach—applying both pretraining and increasing encoder capacity—yields synergistic improvements in mitigating representation collapse.
A section addressing this, or an experiment testing a combined approach, could significantly strengthen the paper’s impact and practical relevance.

2. Discussion on Computational Overheads for Each Collapse:
Increasing encoder capacity is an effective solution for mitigating Embeddings Collapse, but it also increases the computational cost of the model. A discussion on the trade-offs between improved representation quality and increased computational demands would be a valuable addition for practitioners who need to balance performance with efficiency.

3. Discussion on Computational Overheads for Combined Collapse Mitigation:
It would be insightful if the authors explored the computational costs associated with applying both solutions simultaneously. An analysis of how these two solutions interact in terms of performance gains versus computational overheads would enhance the practical relevance of the paper. This discussion would be valuable for practitioners balancing model performance and efficiency.

---

### Official Review · Reviewer_tceG · 2024-10-10
**Representation Collapsing Problems in Vector Quantization**

**Rating:** 7
**Confidence:** 2

**Review:**

I am not familiar with this field, but I like this paper. Here are some advantages:
1. The article is very readable, and the pictures and overall structure are beautiful and reasonable.
2. There is sufficient discussion of related work
3. The contribution seems sufficient, and a detailed analysis is also carried out.

---

### Official Review · Reviewer_yNw6 · 2024-10-11
**Good paper studying timely subject**

**Rating:** 7
**Confidence:** 3

**Review:**

This paper studies vector quantization which is currently widely used in the machine learning community. More specifically, they student the problem of representation collapse when performing vector quantization, and attribute it to two different sources: first, token collapse when a disproportionate number of tokens is allocated to only a few embeddings, and second, embeddings collapse, caused by insufficient encoder parameters. They proposed simple, yet effective solutions to mitigate representation collapse in both cases.

---

### Official Review · Reviewer_efDP · 2024-10-12
**interesting paper, but being out of scope of this workshop**

**Rating:** 2
**Confidence:** 3

**Review:**

This paper is interesting in providing the experimental, comprehensive study of the problem of representations collapsing in vector quantization.
However, even if I personally like this paper, I cannot see any clear connection between this topic and scope of the SafeGenAi Workshop, so the explanation is definitely needed.
Unfortunately, a this point, I can only support rejection from this workshop.